

# Deciphering the preservation of fossil insects: a case study from the Crato Member, Early Cretaceous of Brazil

Gabriel Ladeira Osés[1], Setembrino Petri[2], Bruno Becker-Kerber[3], Guilherme Raffaeli Romero[4], Marcia de Almeida Rizzutto[5], Fabio Rodrigues[6], Douglas Galante[7], Tiago Fiorini da Silva[5], Jessica F. Curado[8], Elidiane Cipriano Rangel[9], Rafael Parra Ribeiro[9] and Mírian Liza Alves Forancelli Pacheco[10]

[1] Programa de Pós-graduação em Geoquímica e Geotectônica, Institute of Geosciences, Universidade de São Paulo, São Paulo, Brazil
[2] Institute of Geosciences, Universidade de São Paulo, São Paulo, Brazil
[3] Programa de Pós-Graduação em Ecologia e Recursos Naturais, Universidade Federal de São Carlos, São Carlos, São Paulo, Brazil
[4] Institute of Geosciences, Universidade Federal do Pará, Belém, Brazil
[5] Institute of Physics, Universidade de São Paulo, São Paulo, Brazil
[6] Department of Fundamental Chemistry/Institute of Chemistry, Universidade de São Paulo, São Paulo, Brazil
[7] Brazilian Synchrotron Light Laboratory, Campinas, Brazil
[8] Department of Physics, Centro Universitário FEI, São Bernardo do Campo, Brazil
[9] Laboratory of Technological Plasmas, Universidade Estadual Paulista, Sorocaba, Brazil
[10] Department of Biology, Universidade Federal de São Carlos, Sorocaba, Brazil

Corresponding author
Gabriel Ladeira Osés,
gabriel.ladeiraoses@gmail.com

## ABSTRACT

Exceptionally well-preserved three-dimensional insects with fine details and even labile tissues are ubiquitous in the Crato Member Konservat Lagerstätte (northeastern Brazil). Here we investigate the preservational pathways which yielded such specimens. We employed high resolution techniques (EDXRF, SR-SXS, SEM, EDS, micro Raman, and PIXE) to understand their fossilisation on mineralogical and geochemical grounds. Pseudomorphs of framboidal pyrite, the dominant fossil microfabric, display size variation when comparing cuticle with inner areas or soft tissues, which we interpret as the result of the balance between ion diffusion rates and nucleation rates of pyrite through the originally decaying carcasses. Furthermore, the mineral fabrics are associated with structures that can be the remains of extracellular polymeric substances (EPS). Geochemical data also point to a concentration of Fe, Zn, and Cu in the fossils in comparison to the embedding rock. Therefore, we consider that biofilms of sulphate reducing bacteria (SRB) had a central role in insect decay and mineralisation. Therefore, we shed light on exceptional preservation of fossils by pyritisation in a Cretaceous limestone lacustrine palaeoenvironment.

## INTRODUCTION

Exceptionally preserved biotas have been recorded since the Precambrian (e.g., *Chen et al., 2014*). They comprise taphonomic windows (Konservat-Lagerstätten of *Seilacher, Reif & Westphal, 1985*), which provide essential evidence for understanding major issues regarding evolution and palaeoecology of ancient ecosystems (e.g., *Raff et al., 2008*). In fact, organisms with low potential of preservation are very promising as taphonomic windows since once they retain fine morphological aspects, this implies in high taxonomic fidelity, representative of an ancient biological community (*Briggs & McMahon, 2016*). The high preservational fidelity of insects from the Crato Member (Santana Formation, northeastern Brazil) defines it as a taphonomic window for an Early Cretaceous ecosystem (*Soares et al., 2013*). Due to this kind of unique record, we know that the evolutionary history of the insects was characterised by major radiation and extinction events in the Cretaceous (*Nicholson, Mayhew & Ross, 2015*), when the diversification of social insects (*Jarzembowski & Ross, 1996*; *Engel, Grimaldi & Krishna, 2007*) and the radiation of flowering plants (*Lidgard & Crane, 1988*) took place. The latter has impacted insect evolution thereafter (*Jarzembowski & Ross, 1996*; *Labandeira, 2014*).

Within the palaeolacustrine setting of the Crato Member, several insect groups display exceptional preservation of non-biomineralised tissues on a micron-scale as well as gross morphological features (*Delgado, 2014*; *Barling et al., 2015*). *Martínez-Delclòs, Briggs & Peñalver (2004)* have pointed out the common association of insect soft-tissue preservation with fine-grained laminated carbonates, which is indeed the case of the Crato Member. Whilst previous studies have considered the preservation of these organisms (*Heimhofer & Martill, 2007*; *Menon & Martill, 2007*; *Delgado et al., 2014*; *Barling et al., 2015*), microtextural and geochemical analyses have not been performed, nor has a detailed taphonomic model been proposed Based on imaging, geochemical, and mineralogical analyses, this paper presents data that supports the central role of microorganisms in the fossilisation of the Crato Member insects. We propose a preservational pathway able to predict interconnections between geobiological and taphonomic processes operating in the exceptional preservation of these insects, which have yielded 3D replicas with mineralised internal soft tissues.

### Geological setting

The fossil insects used in this study are from the Crato Member (Santana Formation, Araripe Basin) located in northeastern Brazil (Fig. 1). It is a continental rift basin, bounded by NE and WNW faults (*Assine, 2007*), formed during the opening of the South Atlantic Ocean (*Brito-Neves, 1990*; *Assine, 2007*).

The base of the Araripe Basin is comprised by the Cariri Formation, proposed by *Beurlen (1962)* (Late Ordovician/Early Devonian) (*Assine, 2007*). Four supersequences are recognised in the Araripe Basin (following *Assine, 2007*): 1- Pre-rift Supersequence: siliciclastic fluvial-lacustrine sediments from both the Brejo Santo and Missão Velha formations, dated to the Late Jurassic by ostracodes and palynomorphs (*Coimbra, Arai & Carreño, 2002*); 2- Rift Supersequence: deltaic, fluvial and lacustrine siliciclastic sediments from the Abaiara Formation, attributed to the Early Cretaceous based mainly

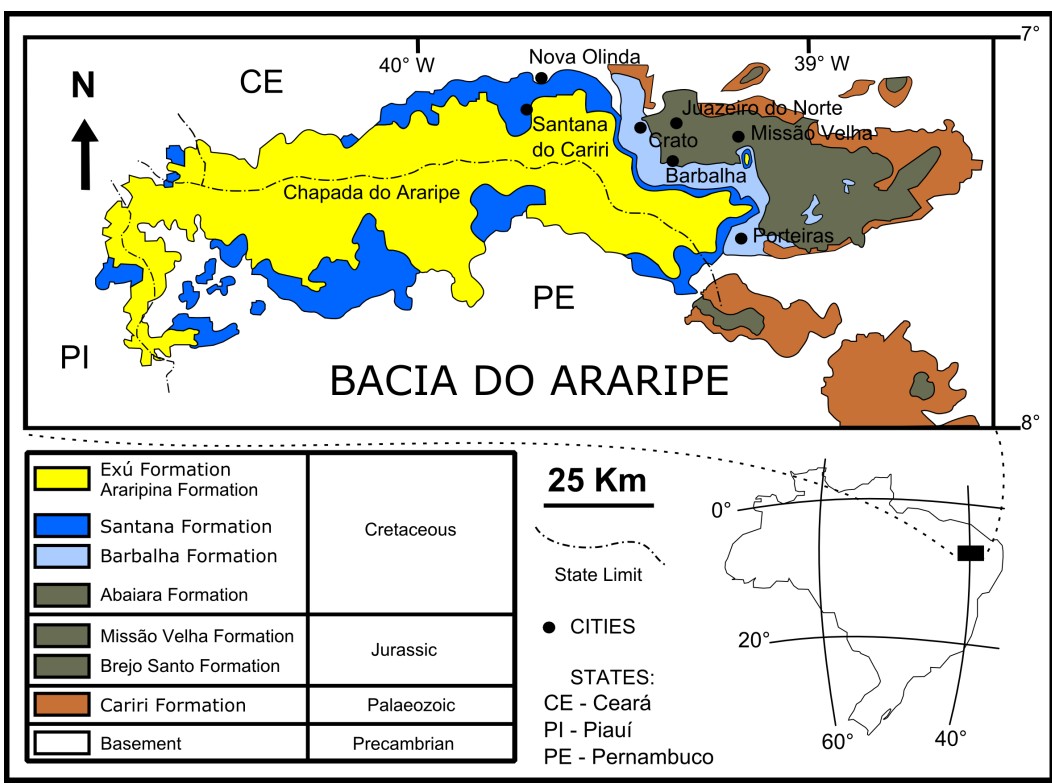

**Figure 1** **Geological setting of the Crato Member.** Geological map of the Araripe Basin, position of the Araripe Basin in the Brazilian territory, and simplified stratigraphic chart of the Araripe Basin. Image credit: modified after *Prado et al. (2016)* (DOI: https://doi.org/10.7717/peerj.1916/fig-1).

on ostracode biozonation (*Coimbra, Arai & Carreño, 2002*); 3- Post-rift Supersequence: Barbalha Formation, with two fluvial (siliciclasts)/lacustrine (pelites and carbonates) cycles and the Santana Formation, both units occurring within the Araripe Group and Aptian-Albian in age (*Coimbra, Arai & Carreño, 2002*). The lower succession of the Barbalha Formation comprises the "Camadas Batateira" which represent the first evidence of an anoxic lacustrine cycle. In the Araripina Formation, heterolitic facies of alluvial fan plains of the Mesoalbian occur. This unit is overlaid by fluvial sediments of the Exu Formation (Araripe Group), located in the top of the Araripe Basin, whose age is uncertain due to the absence of microfossils (*Coimbra, Arai & Carreño, 2002*), although its stratigraphic position suggests an Albian-Cenomanian age (*Coimbra, Arai & Carreño, 2002*; *Assine, 2007*).

The Santana Formation is divided in two members. The Crato Member, the most basal unit, outcrops only in the east portion of the Araripe Basin (*Viana & Neumann, 2000*). Its Late Aptian age is based mainly on palynomorphs (*Coimbra, Arai & Carreño, 2002*). This unit consists of carbonates, forming intermittent banks, more than 20 m thick (*Assine, 2007*). These carbonates are divided into six levels, each one with basal clay-carbonate rhythmites overlaid by micritic laminated limestones, where the fossil insects from this study occur (*Viana & Neumann, 2000*). These lithologies were deposited in a lacustrine

palaeoenvironment. The carbonate levels are interbedded with shales (occasionally rich in organic matter), sandstones, and siltstones (*Viana & Neumann, 2000*).

In the top of the Crato Member, supratidal gypsum layers and shales, known as the "Camadas Ipubi" occur (*Assine, 2007*). Transgressive events led to the deposition of siliciclastic marine sediments, with shales with carbonatic fossiliferous nodules from the Romualdo Member (*Kellner, 2002*), the Santana Formation top stratigraphic unit (*Assine, 2007*). Both "Camadas Ipubi" and Romualdo Member comprise the Late Aptian-Early Albian interval, defined by palynozones (*Coimbra, Arai & Carreño, 2002*). Above the Romualdo Member, a level with marine shell beds occurs, which is covered by regressive freshwater facies (*Beurlen, 1971*), in the upper part of the Santana Formation.

## Materials and Methods

The specimens analysed ("GP/1E") are deposited in the Scientific Palaeontological Collection of the Institute of Geosciences of the University of São Paulo (Brazil). No permits were required for the described study since it was performed after specimens had been deposited in the above mentioned scientific collection. The results herein presented comprise the analyses of the following samples: GP/1E 7105, GP/1E 8440, GP/1E 8397, GP/1E 8827, GP/1E 6820, GP/1E 10368, and GP/1E 9137.

The analyses were made with complementary paleometrical techniques (*Delgado et al., 2014*) on weathered samples, in order to validate the results of several techniques.

Samples were initially observed and photographed in a Zeiss Stemi 2000-C stereomicroscope coupled to a Zeiss AxioCam ICc3 camera. The image acquisition was made in the software AxionVision 4.8.

Micro morphological analyses of the fossil insects were conducted by scanning electron microscopy (SEM) in a JEOL JSM-6010 LA microscope and also in a FEI Quanta 650 FEG microscope, both coupled to an energy dispersive X-ray spectroscopy (EDS) equipment. In the former microscope, an X-ray Dry SD Hyper (EX-94410T1L11) detector with resolution of 129 to 133 eV for the Mn K$\alpha$ line at 3,000 cps was used. To avoid surface charging during SEM inspections samples were coated with a thin layer of gold-palladium using a DESK-V HP Cold Etch/Sputter system. The micrographs were then taken using the secondary electron detector of the microscopes (except one micrograph, which was taken using the backscattered electron detector of the JEOL JSM-6010 LA microscope). All spectroscopic analyses were performed on three main regions of the samples: inside the carcasses, on the cuticle, and on the surrounding rock matrix. EDS point and mapping spectra were employed to highlight qualitative elemental heterogeneities among these three regions. The results obtained with EDS were carefully analysed and interpreted since EDS point analysis may lack spatial representativeness and EDS mapping is a qualitative approach, which may be affected by sample topographic irregularities.

Energy dispersive X-ray fluorescence (EDXRF) analyses were performed for rapid elemental characterisation of heavier elements, previously to EDS in order to select samples to this latter technique. The portable EDXRF equipment consisted of a mini Amptek X-ray tube of Ag anode and a Silicon Drift Detector (SDD - X-ray semiconductor detector) of

125 eV FWHM for the 5.9 keV line of Mn. The measurements were carried out with 30 kV voltage and 5 $\mu$A of tube current and with an excitation/detection time of 100 s.

The quantitative detection of phosphorus in the samples was performed in vacuum, at the soft X-ray spectroscopy (SXS) beamline of the Brazilian Synchrotron Light Laboratory (*Abbate et al., 1999*), following the work of *Leri et al. (2006)*.

The elemental mapping of a whole sample was made by the application of particle induced X-ray emission (PIXE). The analysis was performed in the external beam setup of the 1.7 MV-tandem accelerator of the University of São Paulo. A 2.4 MeV energy proton beam (1 mm in diameter) was used at the sample surface to induce the emission of characteristic X-rays, detected by an AMPTEK XR-100CR (450 $\mu$m thickness, 4.6 mm$^2$ area, 0.5 mil Be-window, and 165 eV energy-resolution at 5.9 keV, and additional X-ray absorber of 300 $\mu$m to avoid high counting rates). The sample was positioned in front of the external beam setup by a robotic sample holder that sequentially moved the sample to cover the fossil area by a matrix of analysed spots (0.7 mm steps in both directions). In each point, the sample holder stands during the detector acquisition time, which in the case of this study was 15 s with a beam current of 10 nA, and saves an X-ray spectra for each point. The maps were created using the peak area (background removed) and the position of each measured point tracked by the robotic sample holder.

The mineralogical composition of both fossils and laminated limestone was analysed by Raman spectroscopy in a confocal micro Raman inVia Renishaw equipment, coupled to a laser of 785 nm wavelength and 300 mW power and a laser of 633 nm wavelength and 17 mW power, and a CCD detector. The Raman spectra were analysed in the software Origin® 8.

## RESULTS AND DISCUSSION

### Microtextural characterisation of the fossils

SEM analysis revealed that fossil exoskeletons (Fig. 2) are preserved by sub-spherical to spherical closely-packed grains, with diameters mainly in the range of 5–10 $\mu$m (Fig. 3A), which are formed by anhedral to euhedral nanocrystals (Figs. 3B–3D). The outer cuticle surface retains fine morphological details (Fig. 3A; *Barling et al., 2015*), built by the close-packing of these grains (Figs. 3A–3D; *Grimes et al., 2002*). The cuticle is also replaced by polygonal lamellar sometimes porous structures likely filled with nanocrystals similar to the ones forming the sub-/spherical grains and with an anhedral microcrystalline mineral phase, with less than 1 $\mu$m (Fig. 3D).

The inner portion of the fossils (Fig. 2) is filled with sub-spherical to spherical generally loosely-packed grains of approximately 1 $\mu$m in diameter, formed by nanocrystals (Figs. 3E and 3F). These grains sometimes have smoothed corroded surfaces and are partially disintegrated or covered by a fuzzy mineral phase (Fig. 3E; as showed by *Barling et al. (2015)* in Fig. 13E). Cuticle-replacing grains have dissolution cavities formerly occupied by crystals, which left empty templates after oxidation (Figs. 3B and 3C ; similar to Figs. 3B and 3D of *MacLean et al. (2008)*). Taking together, such evidence reinforces oxidation.

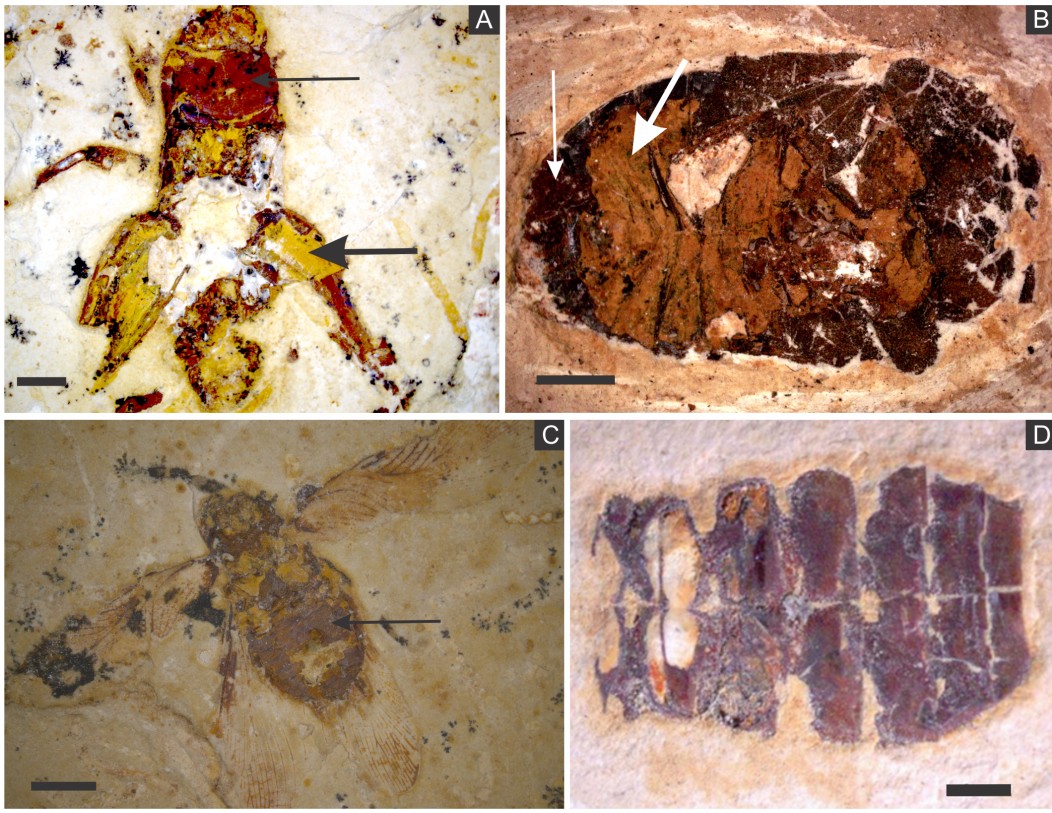

**Figure 2** **(A) orthopteran GP/1E 7105. (B) hemipteran GP/1E 8440. (C) blattodea GP/1E 9137. (D) specimen GP/1E 6820, cuticle of an undetermined insect.** In (A–C), exoskeleton is indicated by narrow arrows and internal part is indicated by wide arrows. The brown, yellow, and orange-brown colours represent the alteration of originally precipitated pyrite (*Barling et al., 2015*). Scale bars = 2 mm (A–C), 1 mm (D). Figure A was modified from *Delgado et al. (2014)*.

In some parts of both cuticle and internal cavities, individual grains are embedded in a smooth matrix, forming clusters that vary in size and shape and are connected by "weblike" structures (Fig. 3F).

## Geochemical analyses

Elemental analyses revealed that iron is more concentrated in fossils than in rock matrix, while calcium and strontium are more concentrated in rock (Figs. 4–6 ; Fig. S1). The preferential distribution of these elements is in accordance with the presence of iron compounds replacing the fossils and the calcitic composition of the rock matrix (*Barling et al., 2015*). Zinc, copper, and lead appear in a higher concentration in the fossils than in the laminated limestone (Figs. 5, 7; Fig. S1). Lead and zinc may be attributed, respectively, to galena and sphalerite (*Heimhofer & Martill, 2007*). Concentrations of copper in fossils may point to the original precipitation of sulphides along with pyrite, galena and sphalerite, reflecting reducing conditions (*Heimhofer & Martill, 2007*).

The low abundance of potassium, aluminium, silicon (Figs. 5 and 6), plus oxygen in the samples can be attributed to an aluminium silicate, probably k-feldspar, which occurs in the laminated limestones (*Heimhofer et al., 2010*), or even to clay minerals formed after

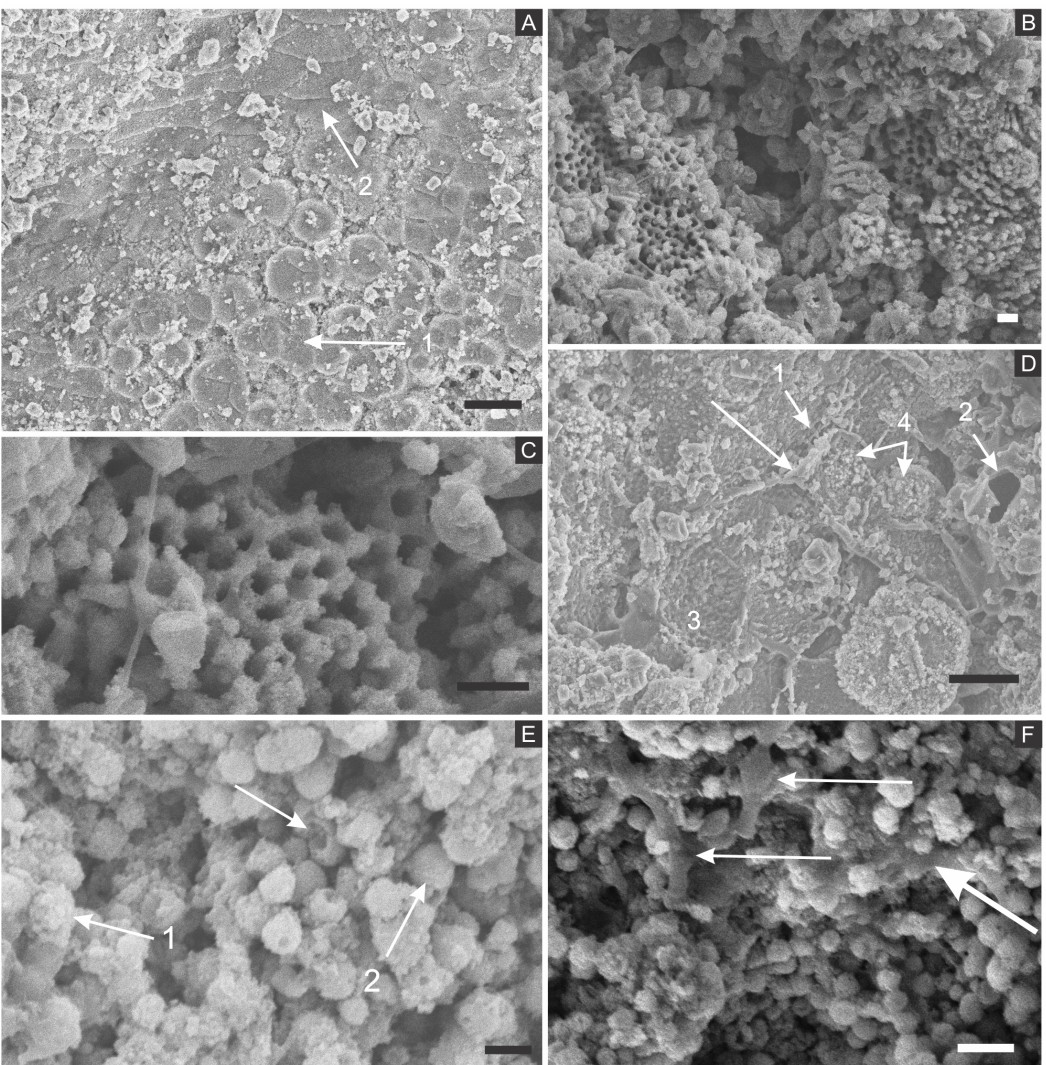

**Figure 3** **Scanning electron microscopy analysis for mineral characterisation of samples.** (A) blattodea GP/1E 9137 (Fig. 2C). Sub-spherical to spherical grains merge (1), yielding a levelled surface (*Grimes et al., 2002*), which retains details of the outer cuticle area (2; e.g., *Barling et al., 2015*). Scale bar = 10 μm. (B) GP/1E 8440 (Fig. 2B). Dissolution cavities delimited by a mineralised template formerly occupied by crystals. Scale bar = 2 μm. (C) GP/1E 8440 (Fig. 2B). Detail of the microtexture depicted in (B). Scale bar = 2 μm. (D) GP/1E 9137 (Fig. 2C). In the cuticle, polygonal structures delimited by lamellae (arrow) occur. These are likely composed by very fine grained pseudomorphs after pyrite. The lamellae are porous in some portions (1 and 2). The polygonal structures are filled with nanocrystals similar to the ones forming the sub-spherical to spherical grains (3) and with anhedral pseudomorphs of microcrystalline pyrite (<1 μm) (4). Scale bar = 10 μm. (E) GP/1E 8397 (Fig. 6A). The microfabrics of the internal cavities are formed by sub-spherical to spherical generally loosely-packed grains (of approximately 1 μm in diameter), formed by nanocrystals (1) and sometimes have smoothed surface (2). The latter is likely an oxidation feature of the former type. The arrow depicts oxidation feature. Scale bar = 1 μm. (F) GP/1E 7105 (Fig. 2A). Some grains infilling internal cavities are embedded in a smooth matrix (wide arrow) and form clusters without a defined shape. "Weblike" structures are indicated by narrow arrows. These features are interpreted as preserved extracellular polymeric substances (EPS). (continued on next page...)
**Figure 3 (...continued)**
Scale bar = 2 μm. (A–F) are secondary electron micrographs. (A) Beam energy: 10 kV, work distance: 11 mm, spot size: 15; (B) Beam energy: 10 kV, work distance: 8 mm, spot size: 15; (C) Beam energy: 10 kV, work distance: 8 mm, spot size: 15; (D) Beam energy: 10 kV, work distance: 11 mm, spot size: 15. (E) Beam energy: 10 kV, work distance: 8 mm, spot size: 15. (F) Beam energy: 10 kV, work distance: 8 mm, spot size: 15.

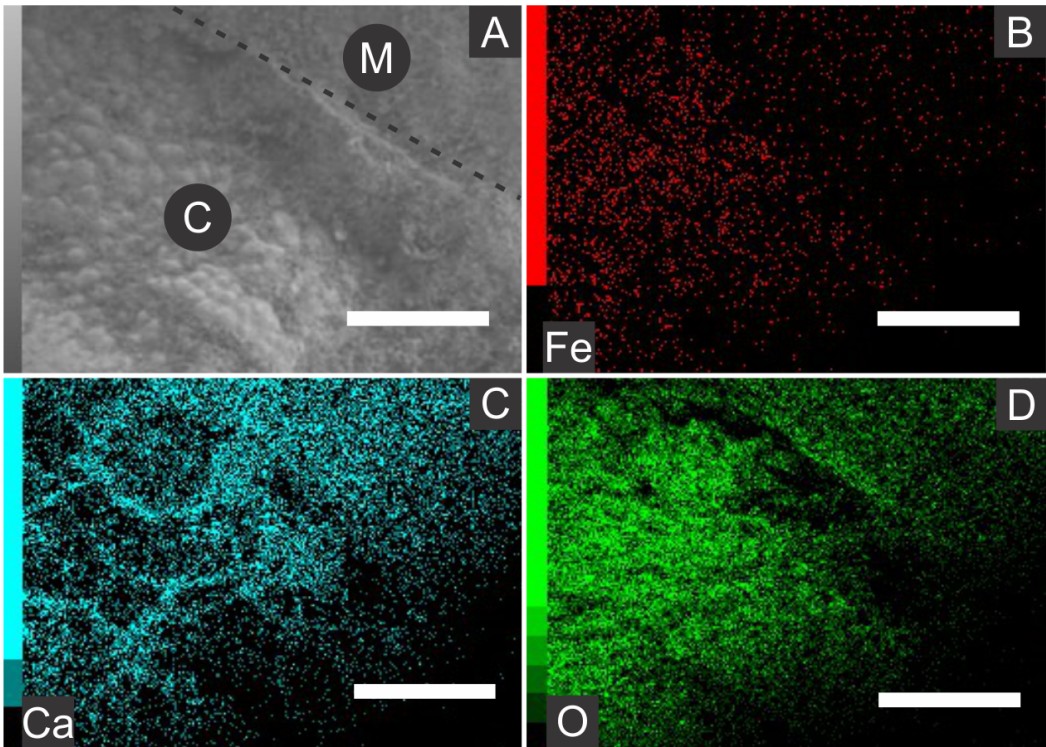

**Figure 4   Energy dispersive X-ray spectroscopy elemental maps of the specimen GP/1E 8440 (Fig. 2B).** (A) Scanning electron microscopy secondary electron micrograph of the specimen (matrix (M) and cuticle (C)). Beam energy: 10 kV, work distance: 8 mm, spot size: 65. (B) iron. (C) calcium. (D) oxygen. Scale bars = 0.5 mm.

feldspar weathering. PIXE mapping of elemental distribution revealed high concentrations of manganese in the rock matrix (Fig. S1), indicating that disseminated pyrolusite does occur (*Heimhofer & Martill, 2007*).

We also showed higher concentrations of phosphorus in the fossils (50.000–60.000 ppm; associated to some areas filled with inner grains (Figs. 5C and 5D), as briefly mentioned by *Delgado et al. (2014)*) than in the limestone (700–800 ppm). The observed positive correlation between the concentration of calcium and phosphorus (Fig. 5) is consistent with the presence of apatite in the samples. EDS elemental mapping of mineral fabrics and "weblike" structure (Fig. 7) revealed a marked preferential concentration of carbon in the latter.

Raman spectroscopy analysis indicated the presence of goethite or amorphous hematite in fossils (Fig. 8; *Faria & Lopes, 2007*). Therefore, iron and oxygen detected by other

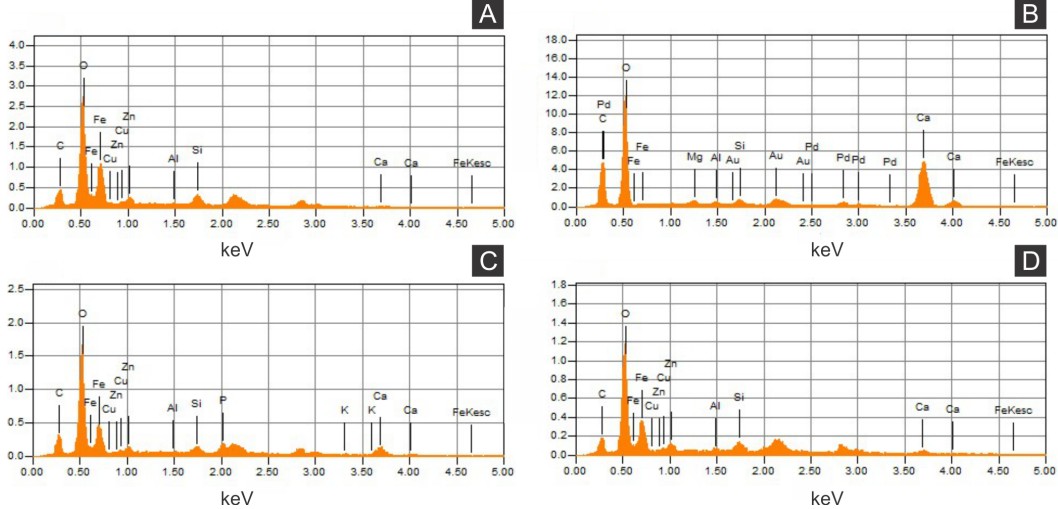

**Figure 5   Energy dispersive X-ray spectroscopy point spectra.** (A) GP/1E 8440 (Fig. 2B). Cuticle. (B) GP/1E 7105 (Fig. 2A). Matrix. (C–D), GP/1E 8397 (Fig. 6A). Internal part of the fossil.

techniques can be associated to these iron oxides/hydroxides, also documented by *Barling et al. (2015)* and *Grimaldi & Maisey (1990)*.

## Preservation of fossil insects

Microcrystals of framboidal pyrite or even of framboid pseudomorphs can be subhedral to anhedral, as for example observed in fresh biofilms (*MacLean et al., 2008*) and replacing Chengjiang (China) *Cricocosmia* worm (*Gabbott et al., 2004*). In samples here analysed, cuticle sub-/spherical grain shape is sometimes obscured, possibly by grain collapse after weathering (*Barling et al., 2015*) although Fig. 6A of *Delgado et al. (2014)* depicts grain shape. Moreover, it is still possible to recognise often regular euhedral to subhedral microcrystal templates (Figs. 3B and 3C ) and anhedral microcrystals (Fig. 6A, inset, of *Delgado et al. 2014*). With such evidence in mind, pyrite framboids can be actually defined as spherical to sub-spherical textures formed by microcrystals often regular in shape and size (*Canfield & Raiswell, 1991*; *Butler & Rickard, 2000*). Therefore, we follow *Delgado et al. (2014)* in their interpretation that cuticle-replacing microfabrics are composed of framboid pseudomorphs, while inner grains are herein considered pseudomorphs after microframboidal pyrite (microframboid *sensu Sawlowicz, 1993*). Indeed, microfabrics are mainly composed of iron and oxygen (Figs. 4 and 5), as also reported by *Delgado et al. (2014)*. Additionally, the polygonal lamellae structures associated with pseudomorphs of pyrite crystals (Fig. 3D) could be interpreted as pyrite overgrowths around originally precipitated pyrite framboids (e.g., Fig. 1D of *Grimes et al., 2002*).

   In comparison to the Crato Member insects, other similarly preserved palaeobiotas include the lacustrine Jehol biota (China) insects, composed of framboids between 6–15 μm (*Wang et al., 2012*), but lacking microframboids. In Crato Member insects, it was possible to differentiate the pyritic microtexture replacing cuticles from that infilling internal cavities or replacing soft tissues (*Delgado et al., 2014*). This difference was not

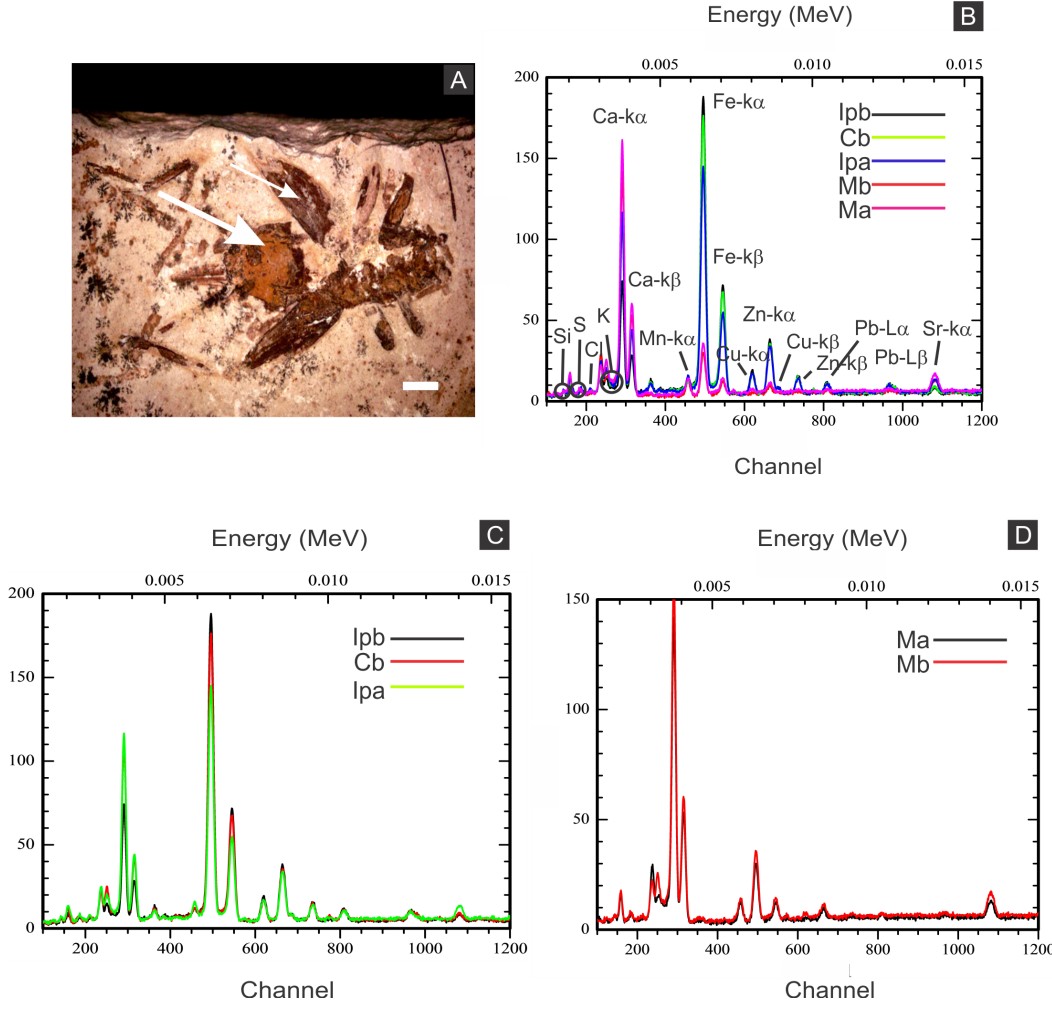

**Figure 6  X-ray fluorescence spectra (EDXRF).** (A) orthopteran GP/1E 8397. Cuticle is indicated by narrow arrow, while internal portion is indicated by wide arrow. Scale bar = 2 mm. (B–D), EDXRF spectra of specimen in A (a) and of specimen GP/1E 8,440 (b) (Fig. 2B). Ip, internal portion; C, cuticle; and M, rock matrix. See (B) for element/peak correlation for all three spectra (B–D).

observed in the Jehol specimens. When cuticle is preserved in these specimens, it is composed of isolated microcrystals (*Wang et al., 2012*), while Crato Member exoskeletons are composed of coarse framboid pseudomorphs. Furthermore, pyritised insects are also found in Cenozoic deposits, including those from (1) the Miocene lacustrine bituminous beds of Rubielos de Mora (Spain), where pseudomorphs after framboidal pyrite also fill the fossils (*Peñalver et al., 1996*) (2) the marine Eocene London Clay (England) (*Allison, 1988a*) and (3) the Eocene Green River Formation (US), where fossils, putatively preserved by iron oxides after pyrite, are hosted by lacustrine calcite mudstones (*Anderson, 2012*).

The geologic record of pyritised insects is less frequent in comparison to other types of mineral replacements, which are mainly restricted to the Cenozoic, and usually yield a higher degree of preservational fidelity than pyritisation. For instance, the Cenozoic insect record includes: silicification (*Palmer, 1957*); phosphatisation and calcification

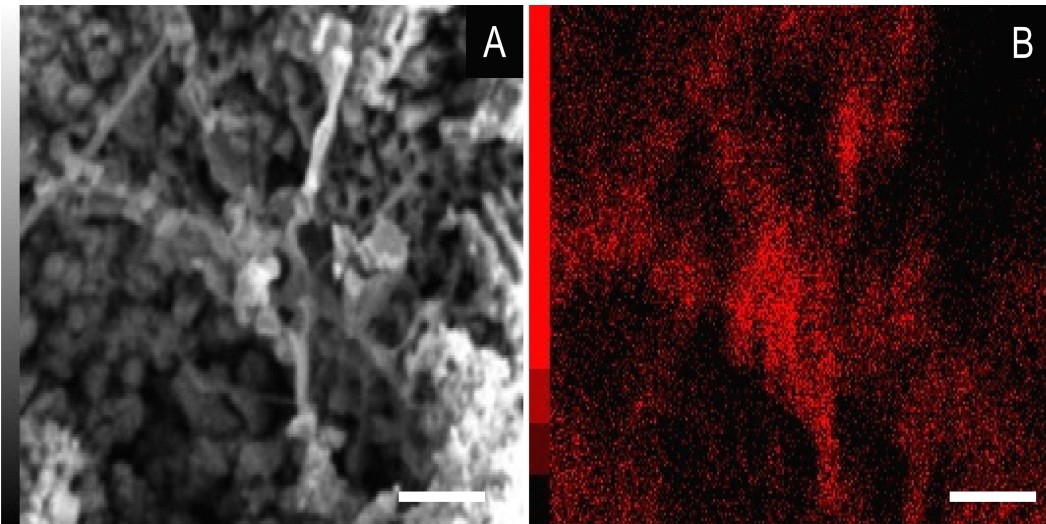

**Figure 7** **Energy dispersive X-ray spectroscopy elemental map of a "weblike" feature.** (A) GP/1E 8827 (Fig. S1). Scanning electron microscopy secondary electron micrograph of the "weblike" (putative preserved extracellular polymeric substances) feature and the surrounding pyrite pseudomorphs. Beam energy: 10 kV, work distance: 11 mm, spot size: 65. (B) Carbon map of the region showed in A. The colour pattern (carbon distribution) may, alternatively, reflect sample topographic irregularities. Scale bars = 5 μm.

(*Leakey, 1952*; *Duncan & Briggs, 1996*; *Duncan, Briggs & Archer, 1998*; *McCobb et al., 1998*; *Grimaldi & Engel, 2005*; *Schwermann et al., 2016*); and specimens preserved within gypsum crystals (*Schlüter, Kohring & Gregor, 2002*) Surprisingly, possibly silicified insects from Geiseltal (Eocene of Germany) have preserved respiratory system (tracheae), subcellular structures of muscles, digestive tract, reproductive organs, and glandular tissues (*Voigt, 1938*).

We consider that the specimens herein studied were pyritised in early diagenesis. This is the most accepted hypothesis for pyritisation in other Konservat Lagerstätte (e.g., *Briggs, Bottrell & Raiswell, 1991*). But other propositions were raised for this kind of fossil preservation. It is possible that iron minerals (framboids and euhedral crystals) occurring in Chengjiang fossils have formed in late diagenesis (cf. *Forchielli et al., 2014*). In order to both rule out the null hypothesis (i.e., late diagenesis pyrite formation) and support the alternative hypothesis (i.e., early genesis hypothesis), we present the following arguments:

1. We present evidence for 3D muscle fibre–a direct replication of soft-tissue micro morphology with high fidelity of detail preserved, as shown below (*Grimaldi, 2003* also depicts muscle fibres; (*Barling et al., 2015*) show preservation of genitalia)— replacement by framboidal pyrite pseudomorphs, which would be quite unexpected in a late diagenetic or even weathering mineral replacement, once, in fact, this kind of high and detailed preservation degree may be obscured by later diagenesis;

2. We consider that it would be quite improbable to insect carcasses remain in 3D until late-stage pyritisation took place, so it should have occurred early. Cuticle is made of large merged pseudomorphs arranged in a way that has even preserved fine morphological details of external cuticle, besides yielding 3D cuticle replicas.

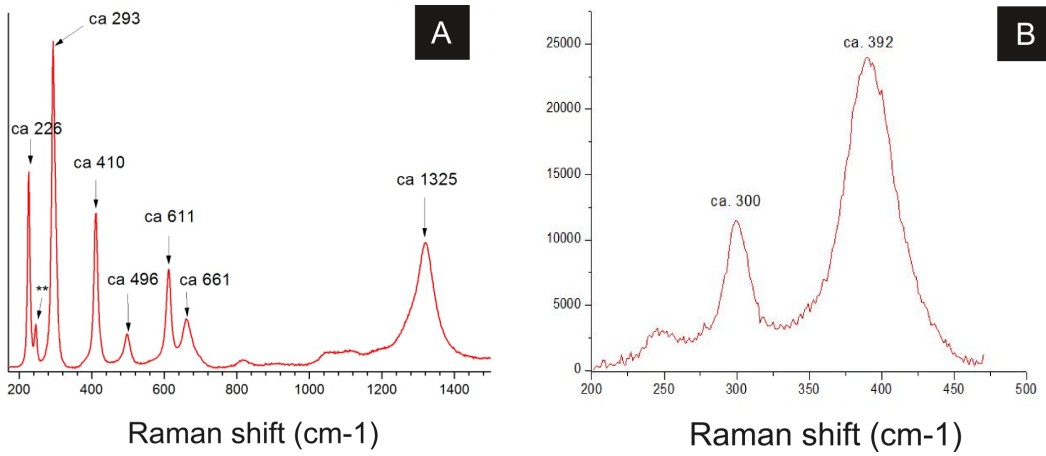

**Figure 8 Raman spectra of insect cuticle.** (A) spectrum of an iron oxide/hydroxide (amorphous hematite or limonite (*Faria & Lopes, 2007*)) of cuticle in Fig. 2D (** = ca 245). (B) spectrum of goethite of the cuticle of the fossil GP/1E 8440 (Fig. 2B). A laser source of 785 nm was used in B and other laser source of 633 nm was used in (A). (A) magnification = 20× , exposure time = 20s, accumulation number = 30, laser power = 1%; (B) magnification = 50× (long working distance), exposure time = 10s, accumulation number = 30, laser power = 0.05%.

Furthermore, fossil tridimensionality also leans on carcass infilling (*Martínez-Delclòs, Briggs & Peñalver, 2004*) by microframboids, a process that must have occurred early to prevent fossil compression (*Peñalver et al., 1996*);

3. We consider that the widespread fossil pyritisation is hardly explained in a moment other than early diagenesis, when the most decay-prone organic matter is still available for SRB. Indeed, mineralisation preferably stabilises labile substrates (*Butterfield, 1995*). Therefore, it is difficult to understand (1) how pyrite would be concentrated in the carcasses, (2) the process which has resulted in framboid size variation along carcasses (discussed below), and finally (3) evidence for preserved extracellular polymeric substances (EPS) deeply associated with microfabrics, if we favour the null hypothesis;

4. Several well-grounded controls for early diagenetic pyritisation were fulfilled in the Crato palaeolake, such as scattered organic matter was low in the sediment (*Neumann et al., 2003*), lack of bioturbation activity, and anoxic conditions (*Heimhofer & Martill, 2007*; *Schiffbauer et al., 2014* and references therein);

5. It is widely demonstrated and accepted that pyrite is precipitated by SRB during early diagenesis, leading to organic matter mineralisation (*Briggs, Bottrell & Raiswell, 1991*; *Briggs et al., 1996*; *Grimes et al., 2002*; *Briggs, 2003*; *Gabbott et al., 2004*; *Schiffbauer et al., 2014*; *Liu, 2016*, among others).

Some of the filamentous structures associated with microfabrics could be interpreted as soft-tissue decay amorphous products, as reported in a taphonomic experiment carried out by *Briggs & Kear (1993)* using decaying shrimps, but they are different from soft tissues preserved in the insects (Fig. 9). Moreover, several observations support that these structures, which seems to have been originally flexible and pliable, are putative remaining fragmentary extracellular polymeric substances (EPS) (Figs. 3F and 7;

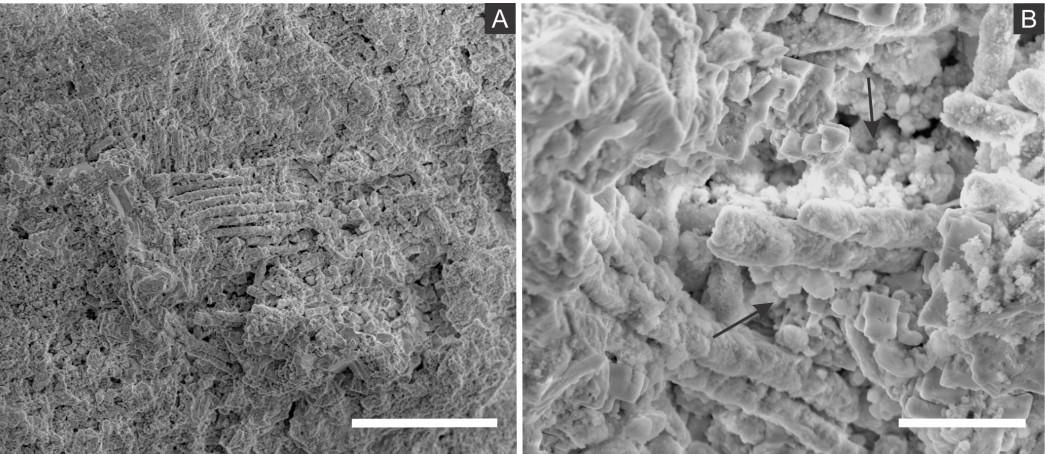

**Figure 9  Scanning electron microscopy micrographs of putative muscular fibres.** (A–B) GP/1E 7105 (Fig. 2A). (A) putative muscular fibres in a broken portion of the cuticle. Scale bar = 50 μm. (B) microfabric (arrows) associated with the putative muscular fibres. Scale bar = 10 μm. (A–B) are secondary electron micrographs. (A) Beam energy: 5.000 kV, spot size: 3.0, work distance: 14.5 mm; (B) Beam energy: 10.00 kV, spot size: 3.0, work distance: 14.4 mm.

e.g., Fig. 10F in *Toporski et al., 2002*; Fig. 3A in *MacLean et al., 2008*; Fig. 3F in *Wang et al., 2012*), and not EPS modern contamination, confirming other current interpretations (e.g., *Delgado et al., 2014*):

1. These structures occur in fossils and were not found in the matrix (*Toporski et al., 2002*), although EPS has been both found associated to calcite and microfossils in the host rock and related to calcite genesis (*Catto et al., 2016*);

2. Figure 7A, for instance, shows that even after SEM vacuum the "weblike" structure has not collapsed, as it would otherwise be expected since samples were not prepared to avoid the collapse of recent hydrated structures (*Défarge et al., 1996*);

3. If these structures were modern contamination, one would expect the presence of bacteria, however it does not happen. This observation is coherent with pyritisation being slower than bacteria decay, thus hindering bacteria preservation. This would be possible by faster mineralisation processes, such as phosphatisation (*Briggs, 2003*; *Briggs et al., 2005*);

4. The structures are structurally organised with mineral fabrics since putative EPS involves them and microfabrics are embedded in a smooth matrix (Fig. 3F), as already mentioned, enabling grain cohesion, accordingly to the EPS definition of *Characklis & Wilderer (1989)*;

5. We actually expect the occurrence of EPS in the context of organomineralisation, such as the precipitation of framboidal pyrite;

6. Finally, the association of high abundance of carbon to EPS (Fig. 7) is also well documented in the Jehol biota fossil insects (*Wang et al., 2012*). *Barling et al. (2015)* suggested that silica halos surrounding and partially covering Crato Member fossil insects might be attributed to preserved bacterial biofilms, although they have not

provided additional morphological and/or geochemical evidence to support this interpretation.

The above discussed presence of pseudomorphs of framboidal pyrite replacing insects, in association with putative EPS, strengthens the hypothesis that biofilm-forming heterotrophic sulphate-reducing bacteria precipitated pyrite, accounting for the preservation of our fossils (*Briggs, 2003*; *Peterson, Lenczewski & Scherer, 2010*; *Wang et al., 2012*; *Delgado et al., 2014*; *Barling et al., 2015*). Indeed, biofilms develop organic templates and suitable chemical microenvironmental conditions for the nucleation, growth and aggregation of pyrite crystals in framboids (*MacLean et al., 2008*). This explains mineral fabrics with empty cavities in the insects originally filled with pyrite crystals, likely outlined by an organic template (Figs. 3B and 3C); very similar to Plate 14, Fig. 15 of *Love, 1965* and to Figs. 3B and 3D of *MacLean et al., 2008*). Additionally, the relationship between decaying organic matter and pyrite growth (*Brock, Parkes & Briggs, 2006*; *Raff et al., 2008*) has already been supported, for instance, by the presence of organic matter in framboids (*MacLean et al., 2008*), and the infilling of microfossils (*Szczepanik & Bak, 2004*) and of vertebrate bones (*Peterson, Lenczewski & Scherer, 2010*) by framboids. Actually, the same happens with the Crato Member insects, thus endorsing the influence of SRB activity to mineralisation during carcass decay. Finally, biofilms create geochemical gradients, controlling ion diffusion rates, directly affecting mineralisation (*Briggs, 2003*; *Peterson, Lenczewski & Scherer, 2010*; *MacLean et al., 2008*; *Raff et al., 2008*)) and, hence promoting active organomineralisation (*sensu Dupraz et al., 2008*). This has already been evidenced by taphonomic experiments with decaying shrimp carcasses (*Sagemann et al., 1999*), which revealed that geochemical gradients are rapidly developed by oxygen and pH decrease, and sulphate reduction is triggered by anaerobic bacterial decay, leading to iron sulphide formation and soft-tissue preservation.

We propose that during early diagenesis, sulphate-reducing bacteria reduced sulphate ($SO_4^{2-}$) to hydrogen sulphide ($H_2S$) (*Heimhofer & Martill, 2007*) and, possibly, ferric iron ($Fe^{3+}$) to ferrous iron ($Fe^{2+}$) (*Coleman et al., 1993*; *Gabbott et al., 2004*; *Popa, Kinkle & Badescu, 2004*; *Heimhofer & Martill, 2007*) dissolved in pore water solutions, leading to pyrite formation, which is generally controlled by the amount of dissolved sulphate, reactive iron minerals and available decay-prone organic matter (*Berner, 1984*; *Skei, 1988*; *Sawlowicz, 1993*). This process led to exoskeleton mineralisation (e.g., *Orr, Briggs & Kearns, 2008*). Moreover, the diffusion of pore water solutions into and through insect carcasses also provided ions for SRB, which in turn infested the insects (*Peñalver et al., 1996*; *Briggs et al., 2005*), mediating the precipitation of minerals, mainly microframboidal pyrite, which covered the internal soft tissues (Figs. 9 and 13E of *Barling et al., 2015*). Microframboidal pyrite also infilled internal cavities (Figs. 3E and 3F) with remaining organic matter derived from partially decayed soft tissues (*Orr, Briggs & Kearns, 2008*; *Pan, Sha & Fürsich, 2014*). Therefore, distinct soft tissues had variable preservational potentials (*Briggs & Kear, 1993*; *Duncan & Briggs, 1996*) and/or fossilisation processes varied along carcasses (*Gabbott et al., 2004*). The preservational process is summarised in Fig. 10.

The occurrence of coarse framboidal pyrite and fine microframboidal pyrite pseudomorphs can be interpreted as the result of the balance between ion (iron and

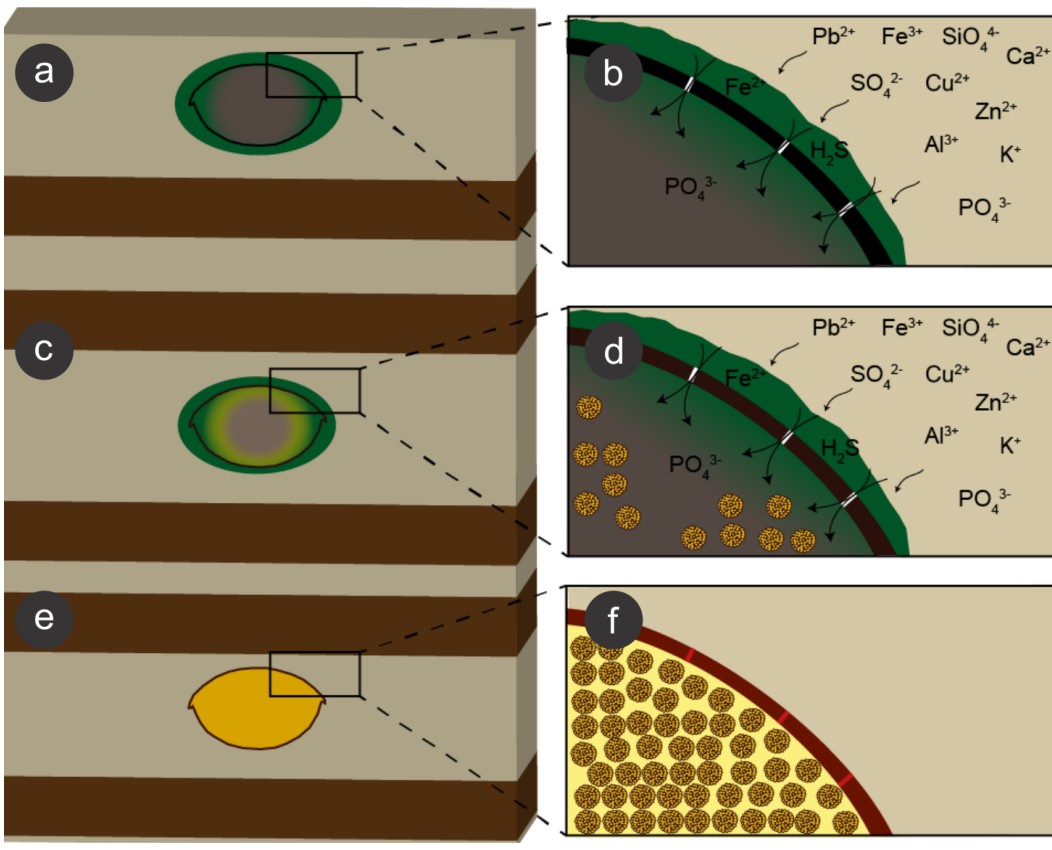

**Figure 10** **Process of preservation of the Crato Member fossil insects.** After final burial (A), ions present in sediment pore water solutions were concentrated in biofilms of sulphate reducing bacteria (SRB) (green) around and within decaying carcasses. Both ions and bacteria entered insects through microcracks (putatively generated by compaction) in the cuticle (black) (B). These bacteria reduced sulphate and, possibly iron (III), resulting in framboidal pyrite formation, which replaced cuticle (brown; C, D). Within the carcasses, labile tissues (grey) were also replaced and replicated (or at least covered) by microframboidal pyrite (C, internal yellow halo and (D), internal yellow spheres). Total carcass collapse was initially avoided by structural strength of both cuticle and internal soft tissues and later prevented by early lithification (*Martínez-Delclòs, Briggs & Peñalver, 2004*) of both exoskeleton and internal soft tissues (E, F), thus yielding three-dimensional replicas. Microcracks were also filled with pyrite (F, red segments in the mineralised cuticle).

sulphate) diffusion and pyrite nucleation rates (*Sagemann et al., 1999*; *Butler & Rickard, 2000*; *Gabbott et al., 2004*). Initially, several pyritic nuclei likely formed owing to an initial high oversaturation of iron and sulphate present in pore water solutions, thus yielding framboids, as proposed for framboid formation in Chengjiang biota fossils and in Jehol biota insects (*Gabbott et al., 2004*; *Ohfuji & Rickard, 2005*; *Wang et al., 2012*; *Schiffbauer et al., 2014*). Moreover, the barrier created by the cuticle, the biofilms (around and inside carcasses), and the already formed authigenic pyrite crystals presumably restricted ion diffusion (lower than nucleation rate) (e.g., *MacLean et al., 2008*) and, thus, also favoured the precipitation of framboidal pyrite, instead of isolated crystals (*Gabbott et al., 2004*). Nevertheless, in comparison to innermost carcass areas, the cuticle received a continuous influx of iron and sulphate from the sediment, which favoured coarse framboid formation,

while finer microframboidal pyrite precipitated within the inner cavities of the carcasses owing to the decreasing influx of iron and sulphate inward. Indeed, initial pyrite saturation and ion diffusion timing can control mineral size (*Sawlowicz, 1993*; *Gabbott et al., 2004*; *Schiffbauer et al., 2014*). Furthermore, the high decay potential of labile internal tissues (e.g., muscles; Fig. 9) also led to an initial increase in $H_2S$ saturation (*Schiffbauer et al., 2014*) and, thus, high nucleation rates and microframboid formation inside the insect carcasses, as suggested by *Gabbott et al. (2004)* to the preservation of the Chengjiang biota. The rapid exhaustion of highly decay-prone organic matter by SRB would limit sulphide production and further crystal growth, accounting for microframboid minute size (*Gabbott et al., 2004*).

Geochemical analyses revealed the preferential concentration of copper, zinc, and lead in fossils in comparison to the surrounding matrix. The different abundance of some elements between fossils and their embedding rock has been extensively attributed to the activity of bacterial biofilms, which envelop decaying carcasses and leads to their mineralisation (*Wilby et al., 1996*; *Toporski et al., 2002*; *Westall et al., 2006*; *Laflamme et al., 2011*). Copper, zinc, and lead are able to bond to organic matter (*Šípková, Száková & tlustoš, 2014*). Alternatively, the preferable association of copper and zinc to the carcasses can be attributed to bacterial activity. Indeed, chitinous substrates buried in sediments are able to remove heavy metals from contaminated environments, a process mediated by bacteria (*Kan et al., 2013*). In this sense, the high chemical affinity of copper and zinc with chitin (*Neugebauer, 1986*) further explains the presence of these metals associated with the insects. Moreover, the adsorption of $Cu^{2+}$ to chitin varies in response to pH gradients (*Gonzalez-Davila & Millero, 1990*), which is controlled by biofilms, as already mentioned. The higher lead concentration in the fossils than in the limestone may be related to the association of this element with iron oxide/hydroxides, as reported in an *Archaeopteryx* sample (*Bergmann et al., 2010*), although no causal relationship has been attributed to explain this preferential association. This may also be explained by SRB activity on and within the insect carcasses yielding authigenic precipitation of galena (*Lambrez et al., 2000*).

Local variations of pH created during anaerobic decay control mineralisation, with acid conditions leading to phosphate precipitation, while higher pH values accounts for carbonate mineralisation (*Briggs & Wilby, 1996*; *Sagemann et al., 1999*). In this vein, other authors suggested that the preservation of internal non-cuticular soft tissues of the Crato Member insects has occurred by phosphatisation (e.g., *Barling et al., 2015*), similarly to the preservation of labile tissues within a Jurassic horseshoe crab (*Briggs et al., 2005*), although direct quantitative evidence has not been revealed until SXS data herein provided. The preferential association of apatite to the fossils also points to microbial activity during fossilisation, as noticed elsewhere (*Briggs et al., 2005*). Only calcium poor continental waters have enough high concentrations of phosphate in solution to enable phosphatisation (*Martínez-Delclòs, Briggs & Peñalver, 2004*), which was not the case of the Crato Member palaeolake. Therefore, alternative sources, such as the decay of organic matter (*Allison, 1988b*; *Briggs, 2003*) might have resulted in a high offer of phosphorus (and phosphate) for fossil insect phosphatisation. This process may have been facilitated

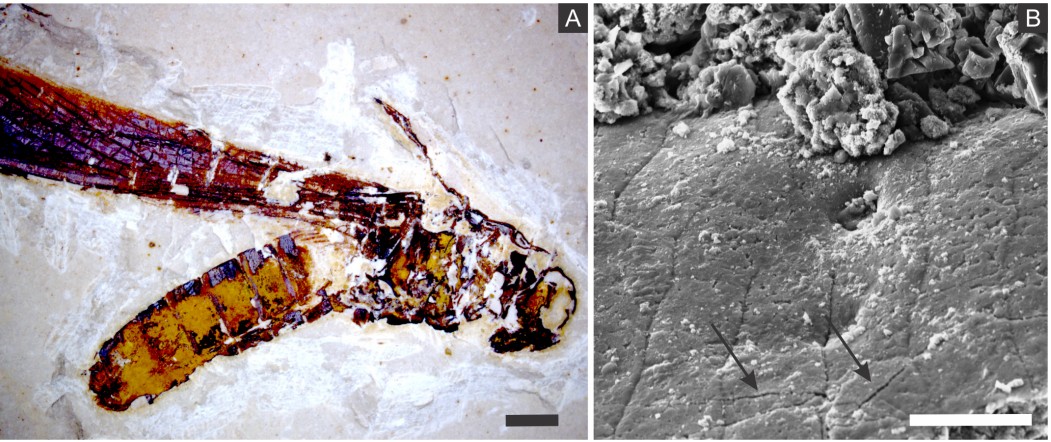

**Figure 11  Microcracks in the cuticle of a specimen.** (A) Orthopteran GP/1E 10368. Scale bar = 2 mm. (B) Scanning electron microscopy secondary electron micrograph showing microcracks in the cuticle (arrows). Scale bar = 10 μm. Beam energy: 10.00 kV, spot size: 3.0, work distance: 11.2 mm.

by the activity of phosphate solubilizing bacteria (*Kan et al., 2013*; *Martínez-Delclòs, Briggs & Peñalver, 2004*).

The diffusion of solutions within decaying carcasses was likely controlled by the lithification rate and, possibly, by exoskeleton microcracks generated by compaction (Figs. 10 and 11). This latter process is an explanation for the preservation of internal tissues in a Jurassic horseshoe crab (*Briggs et al., 2005*), wherein the infestation of bacteria was also facilitated by predation or diseases. Indeed, predation and partial disarticulation of some insects could have facilitated bacteria infestation and the diffusion of ion rich solutions. This mechanism could account for the occurrence of partially disarticulated and fragmented fossil insects, but still with fine details preserved (e.g., (*Barling et al., 2015*)) and with some degree of three-dimensionality due to early mineralisation.

The small size of the microframboidal pyrite pseudomorphs (∼1 μm in diameter) explains the high fidelity of internal soft-tissue preservation (*Briggs, 2003*; *Delgado et al., 2014*), as observed in a taphonomic experiment carried out by *Briggs & Kear (1993)*. We suggest that total carcass collapse was initially prevented by exoskeleton and internal tissue mechanical resistance to compression (*Peñalver et al., 1996*; *Orr, Briggs & Kearns, 2008*; *Pan, Sha & Fürsich, 2014*). Thereafter, further compaction was likely prevented by carcass mineralisation (*Martínez-Delclòs, Briggs & Peñalver, 2004*), yielding three-dimensional insect replicas (Fig. 11), as suggested for the Jehol biota insects (*Wang et al., 2012*; *Pan, Sha & Fürsich, 2014*) and for Miocene insects from Spain (*Peñalver et al., 1996*).

The exceptional preservation of Crato Member insects reflects palaeoenvironmental conditions. Isotopic analyses of carbonate carbon and oxygen performed in the Crato Member basalmost laminated limestones revealed that the depositional palaeoenvironment was a freshwater stratified lake poorly connected with external water sources, with stagnant, anoxic, and at least episodic hypersaline bottom waters (*Heimhofer & Martill, 2007*; *Martill, Loveridge & Heimhofer, 2007*; *Heimhofer et al., 2010*). Water column stratification may have been related to stagnation and/or high rates of surface water primary productivity

providing a high amount of organic matter, the decay of which by aerobic bacteria reduced bottom water oxygen and eventually led to anaerobic conditions in deep waters (*Heimhofer & Martill, 2007*). Furthermore, the occurrence of salt pseudomorphs and xerophytic vegetation pollen supports a semi-arid to arid palaeoclimate (*Heimhofer & Martill, 2007*).

*Melendez et al. (2012)* proposed the influence of photic zone euxinia (PZE) to the preservation of biomarkers and to exceptional fossil preservation (*Heimhofer & Martill, 2007*). The isorenieratene biomarker was reported in the Crato Member laminated limestones by *Heimhofer & Martill (2007)*. This pigment is used by green sulphur bacteria (Chlorobiaceae) in anoxygenic photosynthesis (*Schwark, 2013*). This implies that the palaeoenvironment was, at least, temporarily stratified in relation to $O_2$ and $H_2S$ yielding euxinic photic zone (EPZ), being $H_2S$ likely produced by SRB within the sediment (*Heimhofer & Martill, 2007*). Following this rationale, degradation was diminished after carcasses entered the EPZ since the blockage of autolysis is triggered by reduction and/or anoxic conditions (*Raff et al., 2008*).

*Menon & Martill (2007)* presented data showing that Crato Member insect taxonomic diversity is dominated (around 60%) by groups that may depend on aquatic habitats, such as bugs (Hemiptera), mayflies (Ephemeroptera), dragonflies (Odonata), and flies (Diptera). This pattern may reflect both high number of individuals inhabiting the uppermost freshwater oxygenated waters (*Menon & Martill, 2007*) and/or a taphonomic bias. Aquatic insects would have had the advantage of inhabiting the depositional setting thus facilitating fossilisation, which is in agreement with the preponderance of groups relying on aquatic environments found in carbonate beds (*Martínez-Delclòs, Briggs & Peñalver, 2004*). Moreover, hypersalinity episodes that have affected the Crato palaeolake plus occasional water mixing could have caused poisoning of once freshwater oxygenated shallow waters by $H_2S$, leading to aquatic insect mass mortality (e.g., *Martins-Neto & Gallego, 2006*), thus yielding a significant record of aquatic insects. Additionally, anoxic conditions prevailing in the waterbody would have inhibited macro-scavenger proliferation, facilitating carcass preservation (*Heimhofer & Martill, 2007*).

Moreover, the palaeoenvironmental stratification in respect of oxygen and salinity likely favoured fossil preservation (*Heimhofer et. al., 2007*). The absence of burrowers (together with grazers and scavengers) in the palaeolake owing to its stratification (*Heimhofer & Martill, 2007*; *Menon & Martill, 2007*) accounts for the lack of bioturbation, which have favoured mineralisation. Indeed, diffusion of $O_2$, sediment hydration, and aerobic decay of $C_{org}$ were prevented (*Callow & Brasier, 2009*) resulting in a zone of ionic saturation, heterotrophic anaerobic activity, then yielding the early precipitation of authigenic minerals, like phosphates and pyrite (*Gehling, 1999*; *Callow & Brasier, 2009*; *Laflamme et al., 2011*; G Osés et al., 2016, unpublished data) Similarly, bioturbation was proposed as a control for the pyritisation of insects from the also lacustrine Jehol biota (*Wang et al., 2012*; *Pan, Sha & Fürsich, 2014*). In addition to the lack of bioturbation, the protection of the water-sediment interface against storms likely contributed to substrate anoxia (*Gehling, 1999*; *Heimhofer & Martill, 2007*). Furthermore, the development of SRB biofilms around insect carcasses at the palaeolake bottom, followed by carcass mineralisation, would have been enabled, for instance, by the lack of grazers in the water-sediment interface

(*Menon & Martill, 2007*). Indeed, the importance of microorganisms, of high salinity, and of the lack of scavengers to the preservation of three-dimensional fossil insects was already noticed by *Duncan & Briggs (1996)* for the preservation of Riversleigh (Tertiary, Australia) 3D insects. The role of microbial mats to three-dimensional insect preservation in palaeolakes was then extended to the Jehol biota and the Crato Member (*Wang et al., 2012*; *Barling et al., 2015*).

Nevertheless, the above discussed factors cannot fully explain pyritisation. The Crato Member fossil insects are typically found in laminated limestone facies with a poor content of organic matter (*Neumann et al., 2003*). Jehol biota pyritised insects (*Wang et al., 2012*) and Chengjiang biota pyritised arthropods, sponges, brachiopods, and other organisms (*Gabbott et al., 2004*) are also exclusive to organic-poor lithologies. In this way, the formation of pyrite is concentrated in the carcasses and not widespread within the sediment (*Gabbott et al., 2004*), which, therefore, we extend to the Crato Member (*Martínez-Delclòs, Briggs & Peñalver, 2004*).

The fossil insects from the Crato Member are the first record of these organisms in lacustrine laminated limestones preserved by pyrite without a volcanogenic sediment origin, as it has been suggested for the preservation of the Jehol biota insects (*Wang et al., 2012*; *Pan, Sha & Fürsich, 2014*). These authors argued that iron and sulphur were nourished by volcanic material, deposited at a siliciclastic-bearing lacustrine system. Nevertheless, *Wang et al. (2012)* considered the role of heterotrophic bacteria as central for insect pyritisation, which was put on debate by *Pan, Sha & Fürsich (2014)*. However, for the Crato Member, sulphate was likely provided by evaporites (*Martill, Loveridge & Heimhofer, 2007*). Anyway, despite commonly preserved in continental setting limestones fossil insects are rarely pyritised given the scarcity of sulphate available in such depositional environments (*Martínez-Delclòs, Briggs & Peñalver, 2004*).

Finally, SEM, EDS, EDXRF, PIXE, and Raman analyses (Figs. 3–6, 8, Fig. S1) suggest that the supergene oxidation and/or hydration of pyrite resulted in the formation of iron oxides/hydroxides (*Sawlowicz & Kaye, 2006*; *Menon & Martill, 2007*; *Wang et al., 2012*; *Delgado et al., 2014*; *Pan, Sha & Fürsich, 2014*).

## CONCLUSIONS

The results of imaging and geochemical techniques suggest that Crato Member fossil insects have been preserved by framboidal pyrite. Based on such evidence, we propose that the diffusion of pore water solutions to and through insect carcasses and their envelopment and infestation by bacteria created microenvironmental geochemical conditions which led to the mineralisation (mainly pyritisation) of insect cuticles and internal soft tissues. These geobiological/taphonomic processes have yielded three-dimensional replicas of insects, keeping morphological details of delicate features (e.g., muscle fibres), which can shed light on taxonomy, systematics, and palaeoecology.

Despite of pyrite genesis being ubiquitous, pyritisation of labile tissues is rare and restricted to few examples in the fossil record (e.g., *Briggs, Bottrell & Raiswell, 1991*; *Gabbott et al., 2004*; *Wang et al., 2012*). Indeed, the exceptional preservation of the

Crato Member fossil insects confirms the importance of the following factors to the formation of *Lagerstätten*: early diagenetic precipitation of pyrite (*Gabbott et al., 2004*; *Wang et al., 2012*; *Barling et al., 2015*) under stratified lake conditions with low energy and without bioturbators (*Gehling, 1999*; *Wang et al., 2012*), associated with microbial activity (*Duncan & Briggs, 1996*; *Wang et al., 2012*; *Delgado et al., 2014*; *Barling et al., 2015*; *Catto et al., 2016*) and fine sediments (*Gehling, 1999*) with low organic matter contents (*Neumann et al., 2003*).

## ACKNOWLEDGEMENTS

We thank the Department of Federal Police of Brazil for intelligence service and proactive initiatives which have diminished the illegal international trade of fossils from the Santana Formation, in the last years. These actions have increased the palaeontological collections of Brazilian universities (e.g., University of São Paulo) in thousands of specimens, providing an unprecedented amount of exceptionally well preserved fossils for Brazilian researchers and foreign collaborators, hence contributing to the development of Brazilian palaeontological research. Most of this material is already legally available for research. We are very grateful to the huge effort made by the staff headed by Professor Paulo Eduardo de Oliveira, Professor Juliana Moraes de Leme Basso, and Ivone Cardoso Gonzales, which improved the infrastructure of the Scientific Palaeontological Collection of the Institute of Geosciences from the University of São Paulo (São Paulo, Brazil), enabling proper fossil storage and organisation. We would like to acknowledge the Astrobiology Laboratory (Institute of Astronomy, Geophysics, and Atmospheric Sciences, University of São Paulo, USP) for Raman analyses, the Institute of Physics (USP) for EDXRF analyses the Laboratory of Materials and Ionic Beams (Institute of Physics, USP) for PIXE analyses, the Laboratory of Technological Plasmas (São Paulo State University) and the Brazilian Nanotechnology National Laboratory for having kindly offered their SEM equipment, and finally, the Brazilian Synchrotron Light Laboratory for SXS analyses. We also would like to thank the support of the graduation programs Ecologia e Recursos Naturais (PPGERN) and Biotecnologia e Monitoramento Ambiental (PPGBMA), both from Federal University of São Carlos (UFSCar-São Carlos and UFSCar-Sorocaba) besides the graduation program Geoquímica e Geotectônica, from USP. We are also grateful to Professor Martin David Brasier, *in memorian* for his stimulating and insightful ideas and to Professor Thomas Rich Fairchild for enlightening discussions. We thank the graduate student Gustavo Evangelista Prado for having kindly offered a vectorised version of his geologic map of the Araripe Basin. We also thank Evandro Pereira da Silva for technical support during Raman spectra acquisition, Evelyn Aparecida Mecenero Sanchez for support in figure preparation, and Hugo Silva Pires Junior for skilful sample preparation. We acknowledge Graciela Piñeiro, PeerJ Academic Editor, and the reviewers Bo Wang and Andre Nel, whose comments have improved our original manuscript. We are also thankful for the language revision made by Izabel Maria da Silva Ladeira from English for You (São Paulo, Brazil).

### Funding

The Brazilian Council for Scientific and Technological Development (process 154062/2014-6) provided a master scholarship for the author Gabriel L. Osés. Micro-Raman equipment (Astrobiology Laboratory, Institute of Astronomy, Geophysics, and Atmospheric Sciences, University of São Paulo, USP) update was funded by the São Paulo Research Foundation, FAPESP, project 2012/18936-0. EDXRF equipment (Institute of Physics, USP) acquisition was funded by FAPESP, process 2012/00202-0. The Brazilian Nanotechnology National Laboratory offered their SEM equipment under the Quanta 15068, 16826, 16923, and 18363 proposals. The Brazilian Synchrotron Light Laboratory enabled SXS analyses under the SXS 14563 proposal. The funders had no role in study design, data collection and analysis, decision to publish, or preparation of the manuscript.

### Grant Disclosures

The following grant information was disclosed by the authors:
Brazilian Council for Scientific and Technological Development: 154062/2014-6.
São Paulo Research Foundation, FAPESP: 2012/18936-0.
FAPESP: 2012/00202-0.
The Brazilian Nanotechnology National Laboratory: 15068, 16826, 16923, 18363.
Brazilian Synchrotron Light Laboratory: 14563.

### Competing Interests

The authors declare there are no competing interests.

### Author Contributions

- Gabriel Ladeira Osés conceived and designed the experiments, performed the experiments, analyzed the data, wrote the paper, prepared figures and/or tables, reviewed drafts of the paper.
- Setembrino Petri and Mírian Liza Alves Forancelli Pacheco conceived and designed the experiments, analyzed the data, wrote the paper, reviewed drafts of the paper.
- Bruno Becker-Kerber analyzed the data, wrote the paper, prepared figures and/or tables, reviewed drafts of the paper.
- Guilherme Raffaeli Romero analyzed the data, wrote the paper, reviewed drafts of the paper.
- Marcia de Almeida Rizzutto conceived and designed the experiments, performed the experiments, analyzed the data, contributed reagents/materials/analysis tools, wrote the paper, prepared figures and/or tables, reviewed drafts of the paper.
- Fabio Rodrigues, Douglas Galante and Elidiane Cipriano Rangel conceived and designed the experiments, performed the experiments, analyzed the data, contributed reagents/materials/analysis tools, wrote the paper, reviewed drafts of the paper.
- Tiago Fiorini da Silva conceived and designed the experiments, performed the experiments, analyzed the data, contributed reagents/materials/analysis tools, wrote the paper, prepared figures and/or tables.

- Jessica F. Curado and Rafael Parra Ribeiro conceived and designed the experiments, performed the experiments, analyzed the data, contributed reagents/materials/analysis tools.

## Data Availability

The research in this article did not generate, collect or analyse any raw data or code.

## Supplemental Information

Supplemental information for this article can be found online at http://dx.doi.org/10.7717/peerj.2756#supplemental-information.

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
