# Peer review of "Deciphering the preservation of fossil insects: a case study from the Crato Member, Early Cretaceous of Brazil"

_PeerJ, doi:10.7717/peerj.2756_

## Round 0.1 · original submission · Minor Revisions

Dear authors,

We have now two reviewers that revised your manuscript “Deciphering the preservation of fossil insects: a case study from the Crato member, Early Cretaceous of Brazil”. The reviewers think that this is an interesting contribution to the taphonomy of the Crato Member and deserves to be published.

In my opinion, this manuscript not only contains a very detailed geochemical analysis to understanding the taphonomic features of the Crato Member fossils (which can be applied for other similar Konservat-Lagerstätten), but it also could acknowledge for the existence of a not always clear boundary between biostratinomic and diagenetic processes acting in the preservation of exceptionally delicate, soft-bodied fossils. This manuscript would be almost ready for to be accepted for publication in PeerJ, but please, consider the inclusion of an analysis of the interesting reviewer’s observations in your discussion, and the revision of the suggested references, and resubmit the manuscript. Otherwise, you can send a rebuttal letter explaining your point to refuse them.

With my best regards,
Graciela Piñeiro

·

Basic reporting

No Comments

Experimental design

No Comments

Validity of the findings

No Comments

Additional comments

The Crato Member Lagerstätte is known for the exceptional preservation of various fossils including plants and insects. The taphonomy of this biota remains unclear, and has attracted much attention recently. This paper is a significant contribution to a better understanding of the taphonomy of the biota. The paper is technically sound, with accurate descriptions, sufficient illustrations. I think it is suitable for its publication in PeerJ.

I have only one concern about the pyritization which may form during the later diagenetic or weathering processes. Fossils are variably altered during weathering, and loss of the original preservational characters may occur. For examples, the pyritization has previously been considered an important taphonomic path of the Cambrian Chengjiang fossils. However, Forchielli et al. (2014), based on unweathering and weathering fossils, revealed that pyrite or iron oxides could be formed in the later diagenetic processes. They found “(pyrite or iron oxides) mineral precipitation onto organic carcasses of the Chengjiang-type fossils is mostly confined to later diagenetic processes and thus is not crucial for the early fixation of easily decayable tissues in Burgess Shale-type faunas.” The same process might occur in Crato Member Lagerstätte too. I think the authors should give some evidence to exclude this possibility.

Forchielli et al., 2014. Taphonomic traits of clay-hosted early Cambrian Burgess Shale-type fossil Lagerstätten in South China. Palaeogeography, Palaeoclimatology, Palaeoecology, 398, 59–85.

·

Basic reporting

no comments

Experimental design

no comments

Validity of the findings

no comments

Additional comments

A very interesting paper, that goes much farther in the analyses than previous ones on the same topics. The conclusions about the quality of the water of the paleolake are quite interesting, it could be interesting to discuss about the presence of a rich aquatic insect fauna (bugs, Ephemeroptera, Odonata, flies, etc.). This situation is in fact not very usual because in many paleolakes, aquatic insects are less frequent than terrestrial ones. Maybe it could be put in relation to the poor quality of water at least during some periods, killing the aquatic fauna

Some fossil insects have their physical color preserved in crato formation, despite the replacement of the cuticle by iron
maybe you could say a word about it, but this is optional

you have also some cenozoic outcrops in which muscles are preserved (for instance Geiseltal in Germany), maybe some words about them could be interesting (ref: Voigt, E. 1938. Weichteile an fossilen Insekten aus der eozanen Braunkohle des Geiseltales bei Halle (Saale). Nova Acta Leopoldina Deutschland, (N.F.), 6 (34): 3-38.)

---

## Round 0.2 · Minor Revisions

Dear authors,

I am glad to see that you included the suggestions and comments from the reviewers into the manuscript and vastly improved the manuscript with the new information provided. Thus, as your academic editor, I consider that the manuscript is almost ready for to be accepted for publication in PeerJ. Please, see the attached annotated manuscript and fix the text and References as indicated, and resubmit the manuscript for to be approved.

In addition, I would like to make some comments: I agree that pyritisation should be produced at early stages of diagenesis and share what you explained in your comments to Dr. Wang. The only aspect that may influence that process could be the time of exposition of the carcases after death. In environments such as the suggested by the Crato Member it is very possible that exposition would be extended, as the deposition rate might be very low. Maybe a comment about this factor should be included, particularly if you consider that the preserved carcases were buried faster and you can provide evidence for that. If the exposure could have been long, it could mean that the diagenetic processes (after the definitive burial) would start late after the death of the animals. But it is just a comment, you can use it as you wish.

With my best regards,

Graciela Piñeiro

---

## Round 0.3 · accepted · Accept

Dear Dr. Osés and collaborators,

It is my pleasure to let you know that your manuscript entitled “Deciphering the preservation of fossil insects: a case study from the Crato Member, Early Cretaceous of Brazil“, is now ready for publication in PeerJ. Thank you very much for your open disposition to consider the reviewers and editor suggestions. This will be an important contribution for better understanding of the taphonomic processes that allow the preservation of delicate organisms. I am indeed very glad to have contributed as your Academic Editor, for its publication in PeerJ.

Congratulations!

Yours sincerely,
Graciela Piñeiro